# Investigation on Roles of Packing Density and Water Film Thickness in Synergistic Effects of Slag and Silica Fume

**DOI:** 10.3390/ma15248978

**Published:** 2022-12-15

**Authors:** Yunchuan Zhao, Xuming Dong, Zicun Zhou, Jiangfeng Long, Guoyun Lu, Honggang Lei

**Affiliations:** 1College of Civil Engineering, Taiyuan University of Technology, Taiyuan 030024, China; 2Shanxi Transportation Research Institute Group Co., Ltd., Taiyuan 030032, China; 3School of Materials Science and Engineering, Tongji University, Shanghai 201804, China

**Keywords:** slag, silica fume, cement paste, packing density, water film thickness, fluidity

## Abstract

The ternary blended cement with finer slag and silica fume (SF) could improve the packing density (PD) through the filling effect. The excess water (water more than needed for filling into voids between the cement particles) can be released to improve the fresh properties and densify the microstructure which is beneficial for improving the hardened properties. To verify the hypothesis and reveal how and why (cement + slag + SF) the ternary blends could bring such advantages, the binder pastes incorporating slag and SF with various water-to-binder ratios were produced to determine the PD experimentally. To evaluate the optimum water demand (OWD) for maximum wet density, the influence of the dispersion state of the binder on PD was investigated using the wet packing density approach. The effect of PD of various binary and ternary binder systems on water film thickness (WFT), fluidity, setting time, and compressive strength development of cement paste was also investigated. The results show that the ternary blends could improve the PD and decrease the water film thickness (WFT). The enhanced PD and altered WFT are able to increase fluidity and compressive strength. The ternary blends could improve the compressive strengths by increasing PD and exerting nucleation and pozzolanic effects.

## 1. Introduction

The development of alternative binders for Portland cement has been considered one of the key solutions for reducing CO_2_ emissions, especially in developing countries. There are three motivations to reduce clinker content: (1) ecological benefits, CO_2_ emissions reduction [1], (2) economic benefits, cost reduction [2], and (3) technical benefits, improving the performance of cement and concrete [3]. Compared with other routes [4,5,6], partially substituting supplementary cementitious materials (SCMs) for Portland cement (PC) is one of the most efficient solutions considering the ecological compatibility and technical characteristics [7,8,9,10].

During the concrete preparation, the added water must be sufficient to fill the gaps between solid particles, because any unfilled gap will become air voids, which may greatly reduce the mechanical properties and durability [11]. Then, the addition of extra water (more than the water needed to fill the void) will form a water film on the surface of the solid particles to lubricate the solid surface so that the cement paste reaches the desired fluidity [12]. The packing density (PD) of solid particles determines the volume of voids which is then needed to be filled with water and is an important parameter to control the rheological and the final hardened properties [13]. Under the same water-to-cementitious materials ratio (W/CM, volume ratio), a higher PD requires less water to fill the gap and the excess water could be released to increase the fluidity of cement paste [12]. In addition, it will allow a lower W/CM ratio to increase the mechanical strength of the cement paste with the same flow requirements [14].

To further reduce the W/CM ratio and produce a higher strength, fine particles are usually added to increase PD. Silica fume (SF) with a particle size of 1/10 of cement is commonly used to fill the gap between cement particles to improve the PD of UHPC [15,16]. The W/CM ratio could be reduced to 0.20 or even lower to achieve ultra-high strength. Among various binders [17,18,19], SF is considered to be the most effective solid for filling the void due to its ultra-fine particle size. Previous studies have shown that incorporating 25% SF by volume can increase PD from 0.654 to 0.736 [20]. However, the effects of SF reported by different researchers on the workability of paste/mortar/concrete vary significantly [19,21,22]. Chen [23], and Shi et al. [24] found that the maximum SF addition can reach as high as 10%, which will not significantly reduce the fluidity and viscosity.

Considering the remarkable size difference between cement and SF, it is helpful to fill the void by using other types of particles. Increasing specific surface area (SSA) excessively will decrease the fluidity and even reduce packing density. Mehdipour and Khayat [25] found that increasing SSA from 425 to 1600 m^2^/kg results in an enhancement in packing density from 0.58 to 0.72, while further increasing SSA from 1600 to 2200 m^2^/kg reduces the packing density from 0.72 to 0.62. To improve the PD without decreasing the workability of the binder system, the ground granulated blast furnace slag is normally chosen to prepare a ternary blended system [26,27]. Aïtcin [28] strongly advocated these ternary blends and found an increase in the overall performance. According to this strategy, slag particles can also fill the voids between cement particles, and SF particles can fill the voids between slag particles. This continuous filling could reduce the voids to a greater extent than adding slag or SF alone. More importantly, a smaller SSA is beneficial to improve working performance [21].

Packing density and water film thickness theories may provide a practical and promising way to characterize and explain the fluidity behavior of fresh paste [29]. According to the theory of PD and film thickness, the mixed water preferentially fills the solids gap, then coats the particle surface, and finally exists in the form of free water. Free water content plays a key role in determining the flow performance of fresh paste, which is usually quantified by water film thickness (WFT) [30]. The WFT has been determined as the excess water to solid surface area ratio [31]. Chu et al. [14] investigated the synergistic effect of MK and SF on compressive and flexural strength, where wet packing density and WFT plays an important role. Li et al. [31] studied the combined effects of superplasticizer and WFT on the flowability of cementitious paste.

Motivated by the discussion mentioned above, the originality of this study is to adopt PD and WFT to characterize the coupled effect of solid content and binder dosage on the flow performance of fresh paste and the compressive strength of hardened paste. Additionally, the available information in the literature mainly focuses on the effect of the ternary blended system on the workability, hydration mechanism, mechanical properties, and durability, while the role of extra water was seldom mentioned. To evaluate the extra water, the water film thickness (WFT) of the adhesive paste mixture was determined according to PD testing. A total of nine blended systems (cement + slag + SF) were produced with a constant water-to-binder ratio (W/B, mass ratio). The PD of blended pastes was measured via a wet packing method and compared with the theoretical PD calculation results. Finally, the fresh and hardened properties of the blended pastes were correlated with mixing parameters, to further analyze and develop the mixing design guidelines for ternary blending of cement, slag, and SF which is widely indicated in ultra-high performance concrete (UHPC) preparation. This work would be beneficial for the utilization efficiency of SCMs with various particle sizes.

## 2. Packing Density of Fine Particles

### 2.1. Wet Packing Test

Different from coarse aggregates, the PD determination of cementitious materials is always a challenge. Different from the solid concentration, PD varies with the water to cementitious materials (W/CM, volume ratio) [32]. The solid volume of the cementitious materials *V*_c_ and the volume of the water *V*_w_ in the mold may be worked out from the following equations [32]:(1)$$V_{c}=\frac{M}{\rho_{w}u_{w}+\rho_{c}R_{c}+\rho_{G}R_{G}+\rho_{S}R_{S}}$$
(2)$$V_{w}=u_{w}V_{c}$$

It should be noted that the W/CM is the same as the water ratio *u*_w_, and the W/CM ratios hereafter are referred to by volume. In which, *M* is the total mass of the paste in the mold, and *ρ*_w_ is the density of water, *ρ*_c_, *ρ*_G_ and *ρ*_S_ are the solid densities of cement, slag, SF, and *R*_c_, *R*_G_, and *R*_s_ are the volumetric ratios of cement, slag, SF to the total cementitious materials. The voids ratio (denoted by *u*) is defined as the ratio of the volume of voids, to the solid volume of granular material, which is related to the total volume of the paste in the mold (denoted by *V*).
(3)$$u=\frac{V-V_{c}}{V_{c}}$$

The solid concentration (denoted by ϕ) of the granular material is defined as the ratio of the solid volume to the total bulk volume of the granular material.
(4)$$\phi =\frac{V_{c}}{V}$$

### 2.2. LPM Model

Despite the experimental evaluation, the theoretical calculation of PD of fine particles has been developed by simplifying some parameters. In this study, the PD of blended powders was estimated based on a linear packing model (LPM) [33,34]. The model was proposed from non-deformable aggregates and their linear interaction. Considering the mixture composed of *n* spherical particles *d*_i_ (*i* = 1…*n*) size, the order is from largest size to smallest size. The maximum PD *γ* can be obtained as follows:(5)$$\gamma =\min\left\{\frac{\theta_{i}}{1-(1-\theta_{i})\sum_{j-1}^{i-1}g(i,j)\alpha_{j}-\sum_{j-i+1}^{n}f(i,j)\alpha_{j}}\right\}$$

The *θ*_i_ and *α*_i_ are the PD and volume fraction of the first monomer in the mixture, respectively. Function *g*(*i, j*) and *f*(*i, j*) represent the linear interaction between fine particles and large particles, which are obtained by filling and loosening particles respectively.
(6)$$g(i,j)={\left(1-\frac{d_{j}}{d_{i}}\right)}^{2.0}+0.4\frac{d_{j}}{d_{i}}{\left(1-\frac{d_{j}}{d_{i}}\right)}^{3.7};d_{j}\leq d_{i}$$
(7)$$f(i,j)={\left(1-\frac{d_{i}}{d_{j}}\right)}^{3.3}+2.8\frac{d_{i}}{d_{j}}{\left(1-\frac{d_{i}}{d_{j}}\right)}^{2.7};d_{j}>d_{i}$$

Based on Equations (5)–(7), the maximum PD of particles can be estimated. Both models require the particle size distribution (PSD) to be measured for all constituent materials [33].

### 2.3. Water Film Thickness

Figure 1a–d shows the filling state of the liquid phase between particles. When the liquid phase is not sufficient to fill the voids among the solid particles, the schematic diagram of the mixture state is shown in Figure 1a. Figure 1b shows the state when the liquid phase fills the void among the solid particles, and the liquid-solid ratio is the plastic limit of the paste, which has the maximum PD of particles. As shown in Figure 1c, part of the water is constrained, which will lead to the flocculation of the mixture and increase the actual water consumption. The influence of particle morphology is shown in Figure 1d. Irregular shape particles need to consume part of water to fill the concave surface of particles, which leads to the decrease of PD and the increase of plastic limit. Figure 1e–h shows the WFT under different water content. After reaching the closest packing, the water film can be formed on the particle surface by continuously increasing the liquid-solid ratio [35]. When the WFT is small, the paste still does not have fluidity. When the water film reaches a certain thickness (critical WFT), the paste begins to flow [36]. At this time, the liquid-solid ratio is the liquid limit. Polycarboxylate superplasticizer can reduce the critical WFT (Figure 1h) and improve the fluidity [37].

It is believed that the WFT, cement paste film, and mortar film are the main factors determining the rheological properties of concrete [38]. The traditional concrete mixture designed method according to a close packing is not able to meet the workability requirements [39], and the addition of fine particles can improve the PD of the cementitious material system. Increasing the SSA of particles will lead to a remarkable absorption of a large amount of water on the fine particles [8], which directly decreases the WFT and the fluidity of the paste. Therefore, particle characteristics are important factors to study the rheological properties of cementitious materials.

In a fresh paste, the theoretical WFT can be used to characterize the spacing of particles in the system [40]. The physical meaning of theoretical WFT is the average thickness of water film wrapping the solid particles, and its value is defined as the ratio of extra water (*V*_ew_) to solid surface area (*S*_p_), as shown in Equation (8). The WFT is related to the PD, water consumption, and total surface area of particles, and Equation (9) is used to calculate the additional water volume *V*_ew_ in the slurry [40].
(8)$$WFT=\frac{V_{\mathrm{ew}}}{S_{p}}$$
(9)$$V_{\mathrm{ew}}=V_{W}-V_{s}\times \frac{1-PD}{PD}$$
where the *V*_ew_, *V*_w_ and *V*_s_ [cm^3^] are the volume of additional water, liquid phase in slurry and solid phase in paste, respectively. PD represents the packing density of solid particles, *S*_p_ is the surface area of solid particles.

## 3. Experimental

### 3.1. Materials

The current ternary blended cementitious system investigated in this work consists of ordinary Portland cement (PO 52.5), slag powder (slag), and silica fume (SF). The slag and SF were employed to investigate the effect of particle size on the PD and corresponding optimum water demands of the paste. The photographs of raw materials can be seen in Figure 2. The cement, slag, and SF are characterized by a Blaine-specific surface area of 326.7 m^2^/kg, 641.5 m^2^/kg, and 22,653.2 m^2^/kg, respectively. The relative densities of the cement, slag, and SF were 3.123, 2.893, and 2.232, respectively. Table 1 presents the chemical compositions of the cementitious materials. Figure 3 shows the particle size distributions (PSD) measured by a laser particle size analyzer, and the LS 13320 laser diffraction tester was performed. A single component polycarboxylate (PCE) superplasticizer (commercially available) was used in this study. The solid content of PCE is 20%.

### 3.2. Mix Proportions

The mix proportions of the cementitious materials are presented in Table 2. Nine mixed proportions of pastes were prepared. Neat cement pastes were used as the reference groups. Eight kinds of samples mixed with slag and SF are named about the content of slag and SF, that is, S represents slag and F for SF. Water content affects wet packing density, and its water-to-cementitious materials ratio (W/CM, volume ratio) is described in Section 3.3.1. The water-to-binder ratio (W/B, mass ratio) of other tests depends on the results of Section 4.1 and Section 4.2.

### 3.3. Test Methods

#### 3.3.1. Packing Density

In this work, a method developed by Wong and Kwan [32] was used to determine the PD of pastes incorporating slag and SF, which is called the “wet packing method”. The method involves varying water-to-cementitious material volume ratios (i.e., before and after reaching the optimum water demand). It is necessary to carry out the wet packing tests in a wide range of W/CM to cover the optimum W/CM ratio. The W/CM ratio was set from 0.4–1.2. The detailed procedures of the proposed test method can be found elsewhere [32]. The photos of pastes with three wet packing densities can be seen in Figure 4. According to the PD, the most suitable W/CM ratio was selected for the production of pastes to study the performance at harden stage, as described in the following section.

#### 3.3.2. Fluidity

According to the results of Figure 5 and Figure 6, blended pastes with the dosages (0–50 wt.%) of slag and SF were prepared with a W/CM ratio of 0.7. To ensure uniform mixing to test setting time, fluidity, and compressive strength, PCE with a cement mass of 0.2% was added to the paste. The flow cone (36 mm top diameter, 60 mm bottom diameter, 60 mm height) specified in Chinese standard GB 8077-2012 [41] was filled with paste on a glass plate. After removing the cone vertically, the sample will gradually spread out. The maximum diameter of the spread sample and the maximum width perpendicular to that diameter was measured. The average of these two values is defined as the fluidity value.

#### 3.3.3. Setting Time

The setting times of cement pastes were determined by the Vicat apparatus, as described in the Chinese standard GB/T 1346–2011 [42]. The samples were kept in a moist cabinet without being disturbed. The testing temperature was 20 °C and the relative humidity was 95%.

#### 3.3.4. Compressive Strength

After mixing the mixture with water adequately, the pastes were poured into cubic molds with a size of 20 × 20 × 20 mm. Then, the molds vibrated for 60 s [43]. The fresh paste was stored in a curing room (95% RH, 20 ± 2 °C). The specimens were demolded after 24 h and cured in the curing room until the design age (1, 3, and 28 d) to test compressive strength. The test procedure for compressive strength was measured according to ASTM C109 [44].

## 4. Results and Discussion

### 4.1. Packing Density and Voids Ratio

Through wet packing tests, the total mass *M* of fresh paste can be obtained, and the voids ratio of the paste can be obtained by substituting it into Equations (1)–(3). Figure 5 shows the voids ratio under different W/CM ratios. For the same group of powders, water consumption also affects the wet PD of powders. When W/CM is 0.6–0.7, the void among the powder reaches the smallest value, that is, it has the largest wet PD.

The wet PD of blended pastes can be calculated by Equation (4). The theoretical PD of the LPM model in Section 2.2 was calculated by MATLAB. All PD results are shown in Figure 6. The difference between the wet PD and LPM model is that the former uses water mixing, while the latter model was assumed at a thorough dry condition. Because water fills in the void of powder particles and occupies the position of powder, the wet PD of the powder is lower than the dry PD. Furthermore, with the increase in water volume, the obtained wet PD decreases. The F10 group contains 10% of SF, and its PD is significantly higher than that of the adjacent group, which is attributed to the fine-grained distribution of SF. Similarly, increasing the amount of SF is beneficial to improving PD.

### 4.2. Water Film Thickness

Figure 7 is the average WFT of solid particles with different W/CM (volume ratio). As shown in Figure 5, the decrease in the W/CM volume ratio makes the WFT decrease gradually. When the water is filled in the void among particles and adding extra water will adhere to the particle surface to form a water film. The WFT has a relationship with PD and the SSA of powder, and the WFT will directly affect the fluidity of the paste.

To explore factors influencing the fluidity, the WFT of particles was calculated when the W/CM ratio is 0.7. In Figure 7, it can be found that the WFT of control group R has the largest value, followed by S20 and S40, and the least WFT is F10Sx. The results show that slag has little effect on the WFT of particles, and the content of SF will directly decrease the WFT of particles, which affects the fluidity.

### 4.3. Fluidity

Figure 8 shows the fluidity of different binder pastes. The fluidity of blended paste is directly related to the content of SF. When 5–10% SF is added, the fluidity of the paste decreases rapidly. This is because the SF particles are finer than cement, and large SSA increases the water demand, resulting in a decrease in the WFT. This led to increasing friction among particles, thereby reducing the fluidity [12]. On the one hand, due to Van der Waals force and electrostatic interactions, the ultrafine particles can easily agglomerate [45], which harms the fluidity. There is a similar reason for decreasing the fluidity caused by the slag. On the other hand, the irregular shape of the slag is not conducive to solid particles rolling (Figure 1d).

As shown in Figure 6 and Figure 8, it can be found that the voids among solids and the amount of filling water decreases, and the WFT of the particles increases due to the PD increases, thereby increasing the fluidity of pastes. However, too many fine SF particles need more water to wet, which increases the viscosity of the cementitious system [46]. The particles cannot be arranged in the most closely packed manner, and the SF particles cannot work as a role of physical filling to replace the water in the void so that the fluidity decreases significantly [47]. It can be found that the test results of fluidity are highly consistent with the calculated results of theoretical WFT (in Figure 7).

### 4.4. Setting Time

Table 3 shows the effect of PD on the setting time of blended paste. As shown in Table 3, the initial setting time is slightly shortened by adding SF, and the final setting time is prolonged by adding 10% SF. This is due to the strong water absorption of SF, which reduces the free water of the paste. The pastes lose fluidity, and the initial setting time is shortened. Meanwhile, the high activity of SF also has a great influence on the setting time [48]. The prolongation of the final setting time may also be related to the deterioration of paste fluidity caused by SF. When the content of slag increases from 20% to 40%, the setting time is slightly prolonged. This is because the cement clinker content reduces, the overall hydration is slowed down and the setting time is prolonged. On the other hand, the pozzolanic reaction is slow, which prolongs the final setting time.

At the same time, SF and slag particles absorb water which could reduce the water between cement particles and retard the hydration. The hydration rate is controlled by gradually released water among hydration. Slag and SF particles fill in the voids between the cement particles so that the distance between cement particles becomes larger. The blended materials content increases and the setting time is prolonged [49].

Compared with the group containing slag, the combination of slag and SF can shorten the setting time, which may be related to the larger PD. At the replacement rate of 20%, the PD of F5S15 and F10S10 is higher than that of S20. The fewer voids of the blended binders will facilitate the rapid filling of the hydration products. The distance between the powder particles gets smaller, which facilitates the formation of a skeleton structure and shortens the setting time [50].

Due to the small amount of water required for the pozzolanic reaction [51], the effective w/c ratio gradually increases with the increase of slag and SF substitution rate. Many studies have found that the w/c ratio has a great effect on the setting time [48,52,53], which is mainly related to the change in paste fluidity. Since the fluidity is highly determined by PD, the PD has a certain influence on the setting time.

### 4.5. Compressive Strength

To further study the effect of PD on the mechanical strength of blended pastes, the compressive strengths determined at 1 d, 3 d, and 28 d are shown in Figure 9. It is noted that the sole addition of 5% SF increases the compressive strength to 74.7 MPa and the sole addition of 10% SF increases the compressive strength to 75.9 MPa, whereas the sole addition of 20% slag increases the compressive strength to 77.2 MPa. The beneficial contribution to the compressive strength of SF and slag may be attributed to its pozzolanic effect and improved PD [54]. When 5% SF and 15% slag are added, the PD reaches 0.656 and the compressive strength reaches a maximum value of 84.4 MPa. Hence, the ternary mixture of slag and SF was added at the same time, rather than the binary mixture of only slag or SF, which may produce a binder paste with the desired PD and compressive strength. When the content of blended materials is less than 20%, the PD and the compressive strength increase. When the replacement percentage of the blended materials is 40%, there is a slight decrease in the compressive strength. The increase in PD is beneficial to the development of strength, which means that the smaller particle voids can be filled by hydration products [55]. Meanwhile, the distance between particles is small, and it is easy to form a skeleton structure [56], which makes the strength develop rapidly and the compactness increase.

### 4.6. Discussion

When slag and SF are added as fillers, the PD increases and excessive water can be released. If the increase of excessive water is more prominent than the increase of solid surface area, WFT increases, otherwise, WFT decreases [21]. As shown in Figure 10, at a low (slag + SF) replacement level, the WFT tends to increase due to the improvement of PD. At a high (slag + SF) replacement level, the PD improvement caused by further adding slag and/or SF became less remarkable. With increasing the fine slag and SF, the PD and solid surface area continues to increase and the WFT tends to decrease.

To examine the roles of WFT in static fluidity, the fluidity results are plotted against the WFT for various PDs in Figure 11. Adding SF can increase PD significantly (Figure 6) but does not always lead to a greater WFT (Figure 7). It can be found that adding too much SF will lead to a decrease in fluidity (Figure 8) because the effect of the increased surface area is more prominent and the WFT decreases significantly (Figure 7). Whether the WFT would increase or decrease is dependent on the relative magnitudes of the percentage increase in excess water ratio and the percentage increase in a specific surface area [57]. As shown in Figure 11, the linear relationship between fluidity and WFT reveals that the addition of slag and SF mainly affects the static fluidity through the corresponding changes in WFT, while WFT and PD increase first and then decrease as shown in Figure 10. Therefore, to balance the high PD and large WFT, adding an appropriate ratio of ternary blending of slag and SF may be the best solution.

The addition of slag and SF is beneficial to obtain a higher PD, a reduction potential of bleeding and a close particle accumulation, which will lead to an increase in the hydration degree of cement [25]. On the one hand, the slag and SF not only have a better filler effect, but also promote the pozzolanic reaction [58], and ultimately improve the long-term strength of cementitious materials (Figure 9). On the other hand, the compressive strengths are higher than that produced from either of their sole use, indicating the synergistic effect of slag and SF [59].

In general, increasing PD without excessively increasing the SSA of particles will increase WFT, thereby increasing static fluidity. The increased PD will reduce water adsorption and permeability [21]. Slag and SF have a certain synergistic effect on the various properties of the blended paste, so the optimal content must be determined according to the ideal performance that must be achieved simultaneously.

## 5. Conclusions

Through a comprehensive analysis of packing density, water film thickness, fluidity, setting time, and mechanical strength, the effects of supplementary cementitious materials with different sizes were evaluated to increase the utilization efficiency. Based on the results of this study, the following main conclusions were drawn:(1)The mixture has the largest wet PD when the W/CM ratio is 0.6–0.7 by wet PD test. The LPM model was used to calculate the theoretical PD and it was found that adding slag and silica fume can increase the PD, and the PD of F5S15 and F10S10 is larger.(2)By calculating the theoretical water film thickness of particles, it was found that the WFT of R is the largest, followed by S20 and S40, followed by F5Sx, and the smallest is F10Sx. The slag has little effect on the WFT of the particles, and the content of SF will directly affect the WFT of the particles.(3)The fluidity of mixtures containing 10% SF or 40% slag decreased significantly. The negative effect of SF on fluidity is more significant than that of slag. The addition of slag and silica fume increases the PD synergistically and improves the fluidity of the paste. F5S35 has the highest fluidity and F10 has the lowest fluidity. The flow spread diameter of fresh paste increases linearly with the increase in WFT, regardless of the influence of binder dosages and solid contents. A larger WFT provides better lubrication to increase the flowability. The addition of SF shortens the initial setting time, the slag will prolong the final setting time, and increasing PD is beneficial to shorten the setting time.(4)The SF has little effect on the early compressive strength, the slag reduces the early compressive strength, and both SF and slag promote the development of the later compressive strength. The PD of F5S15 reaches 0.6648, and the compressive strength reaches a maximum of 84.4 MPa. The blended materials improve the compressive strength by increasing the PD and exerting the nucleation effect and pozzolanic effect. Overall, the enhanced PD and altered WFT increased the workability and compressive strength. It seems that designing a ternary binder with suitable SCMs by considering the particle packing could compensate for the strength loss and generate equal performance while reducing the cement content.

The results of this study will contribute to a further understanding of the flow behavior of fresh paste under various solid contents and binder dosages, which is of great importance to designing mixtures for efficiently and economically.

## Figures and Tables

**Figure 1 materials-15-08978-f001:**
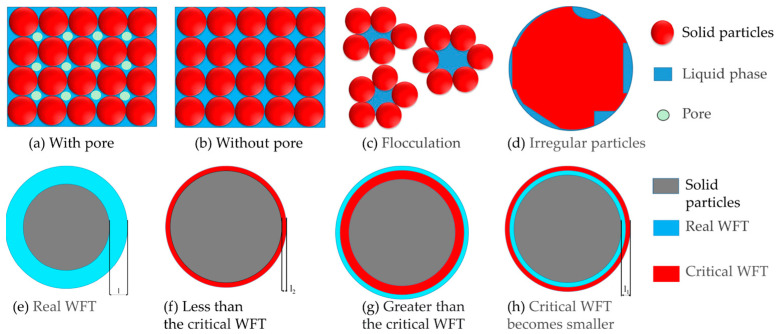
The schematic diagram of the filling state of the liquid phase between particles (**a**–**d**) and water film (**e**–**h**).

**Figure 2 materials-15-08978-f002:**
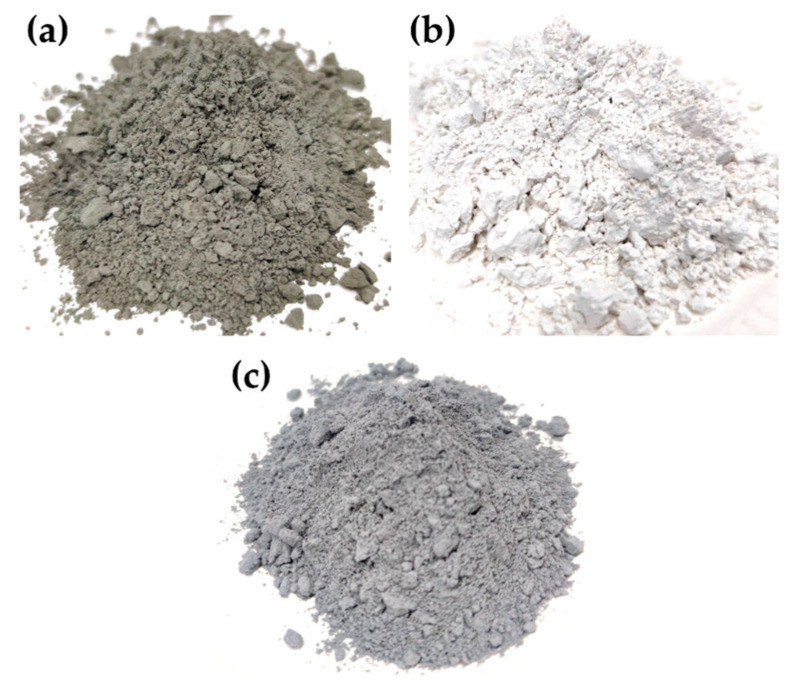
The photographs of (**a**) cement, (**b**) slag, and (**c**) SF.

**Figure 3 materials-15-08978-f003:**
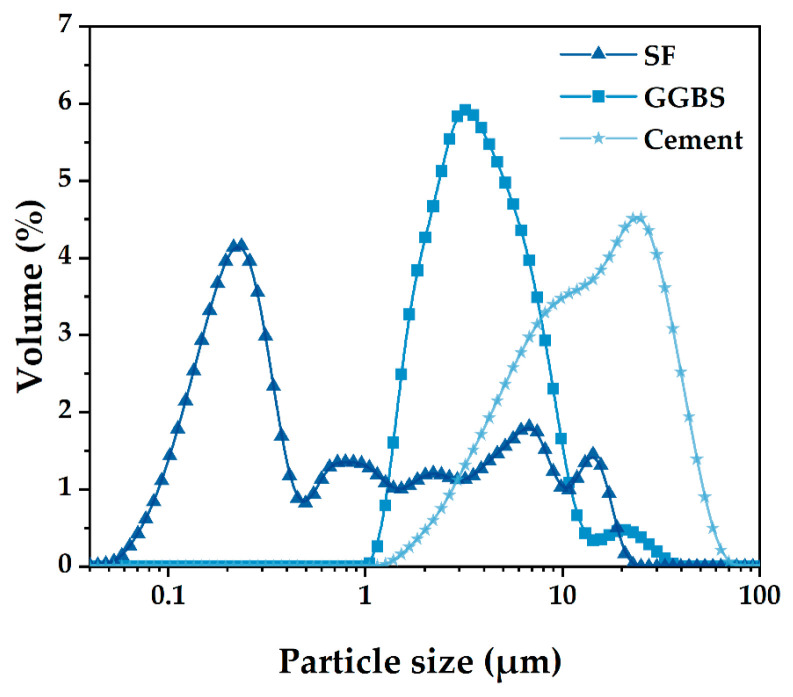
Particle size distribution of cement, slag and SF.

**Figure 4 materials-15-08978-f004:**
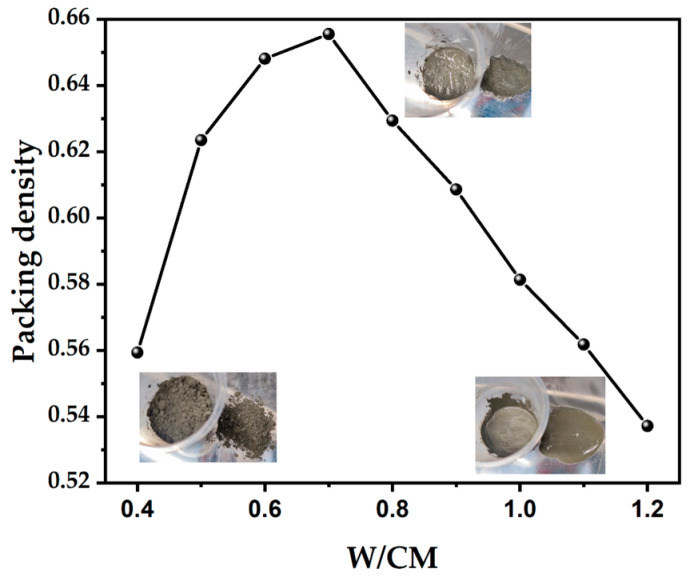
A schematic diagram of the applied wet packing density method.

**Figure 5 materials-15-08978-f005:**
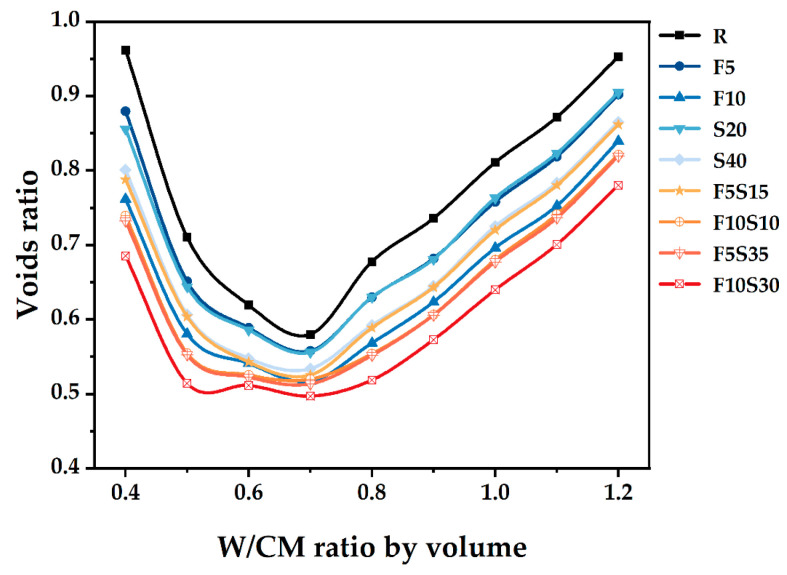
The voids ratio versus the W/CM ratio by volume in relation to the binder compositions.

**Figure 6 materials-15-08978-f006:**
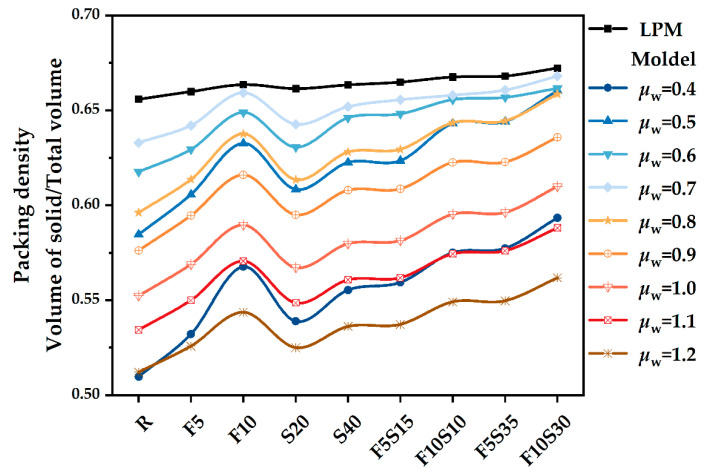
The packing density of binders paste containing OPC, slag, and SF.

**Figure 7 materials-15-08978-f007:**
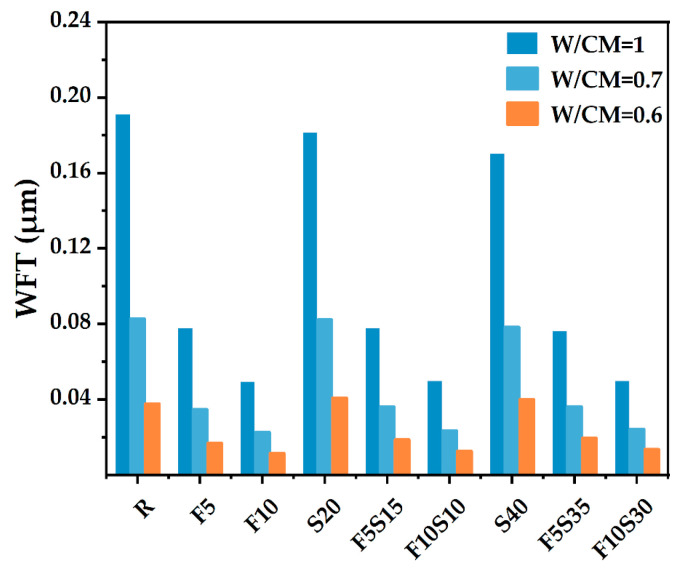
The calculated water film thickness in relation to the W/CM ratios and binder.

**Figure 8 materials-15-08978-f008:**
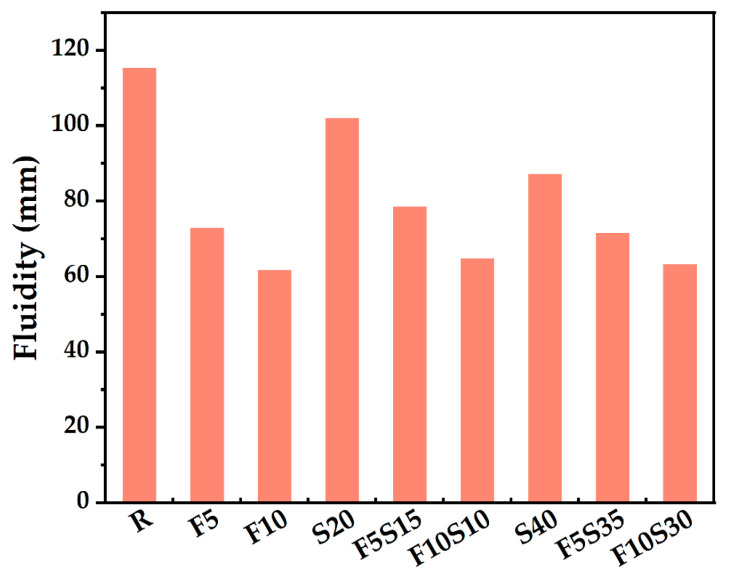
Effect of binder compositions on the fluidity.

**Figure 9 materials-15-08978-f009:**
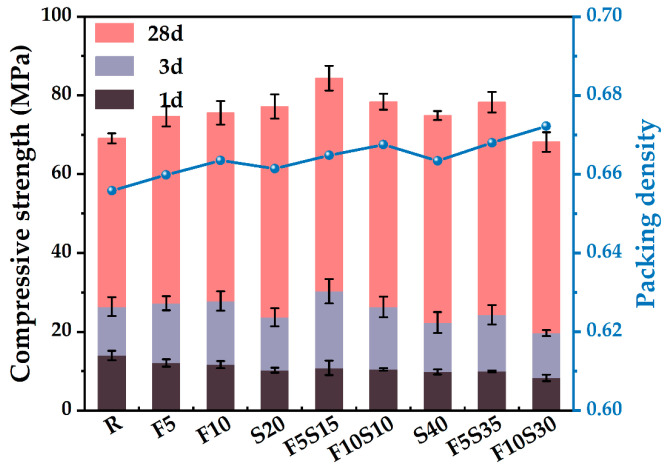
Effect of binder compositions on the compressive strength and packing density.

**Figure 10 materials-15-08978-f010:**
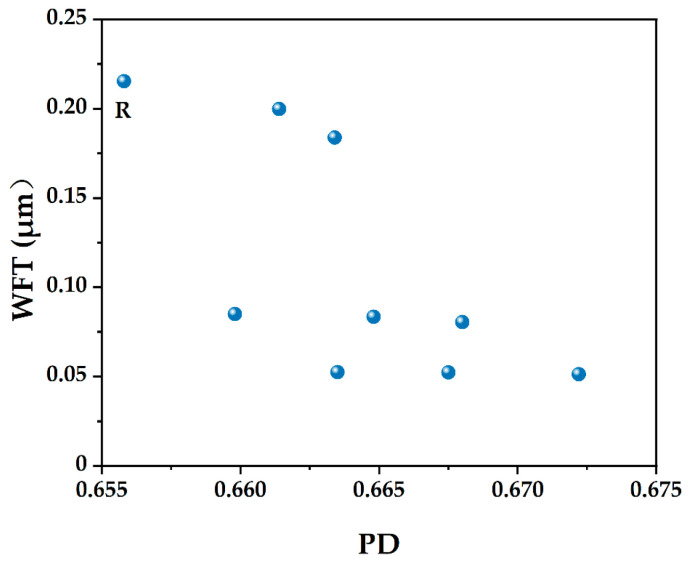
The relation between water film thickness and packing density.

**Figure 11 materials-15-08978-f011:**
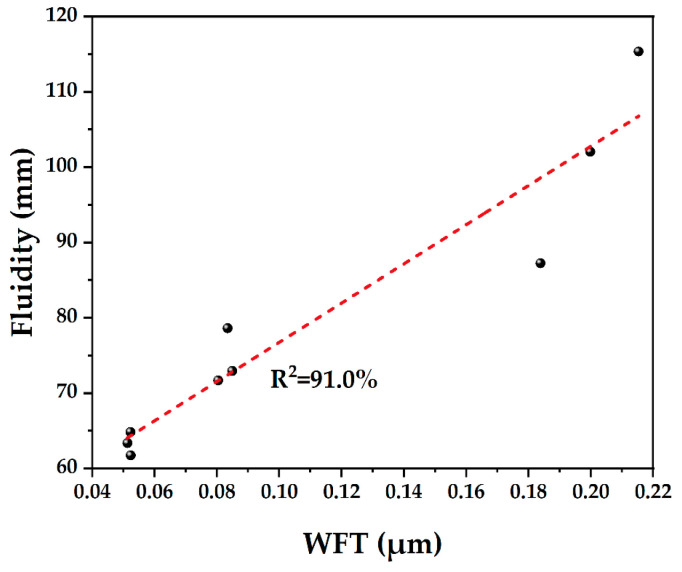
Fluidity versus water film thickness.

**Table 1 materials-15-08978-t001:** Chemical compositions of cement, slag, and silica fume.

Compositions	CaO	SiO_2_	Al_2_O_3_	Fe_2_O_3_	MgO	SO_3_	LOI
OPC	65.00	20.90	4.56	3.23	0.65	2.65	3.01
slag	44.05	32.95	14.46	0.61	5.45	0.63	1.85
SF	0.16	98.23	0.32	-	0.12	0.62	0.55

**Table 2 materials-15-08978-t002:** Mix proportions of cementitious materials.

Groups	Cement	Silica Fume	Slag
R	100	0	0
F5	95	5	0
F10	90	10	0
S20	80	0	20
S40	60	0	40
F5S15	80	5	15
F10S10	80	10	10
F5S35	60	5	35
F10S30	60	10	30

**Table 3 materials-15-08978-t003:** Effect of binders on the setting time (min).

Group	Initial Setting Time	Final Setting Time	Time Difference
R	125	190	65
F5	110	190	80
F10	120	200	80
S20	140	215	75
F5S15	120	200	80
F10S10	130	220	90
S40	150	240	90
F5S35	130	225	95
F10S30	145	250	105

## Data Availability

The data presented in this study are available on request from the corresponding author.

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
