# Peer review of "Investigation on Roles of Packing Density and Water Film Thickness in Synergistic Effects of Slag and Silica Fume"

_materials, 2022, doi:10.3390/ma15248978_

Round 1

Reviewer 1 Report (Previous Reviewer 2)

A well written good paper, issues need to be address are as follows:

1. there are too many "Error! Reference source not found" messages. it means automatic number of figures, tables, equations etc were not controlled properly.

2. Figure 2, particle size distribution is not properly visible, kind make it prominent. is it possible for the authors to present some photographs of SF, cement and slab? (it will give added value to the paper)

3. in all tests, it is very important to present Figure of each test (while testing it in the lab) as said before it will be adding extra value to the results

Many thanks 

Author Response

Please see the attachment. Thanks!

Reviewer 2 Report (New Reviewer)

This study aims to do investigation on roles of packing density and water film thickness in the synergistic effects of slag and silica fume. The paper is interesting and could be accepted for publication after a major revision. So, the authors are invited to address the following comments carefully.

1-      The abstract need to be improved significantly. The authors should discuss their study in more detail. Also, the most important results should be presented at the end of the abstract

2-      A list of notions is recommended to be provided.

3-      The paper needs to polish in terms of English.

4-      The introduction is so short. So, more studies should be discussed in detail to show the gap in previous studies that resulted in doing current investigation

5-      This is strongly recommended to provide a “research significance” section to talk about the novelty of this study in detail

6-      The chemical and mechanical characteristics of materials should be provided

7-      “Error! Reference source not found. ~Error! Reference source not found” in Line 100 is not clear

8-      There are many “Error! Reference source not found.” That the reason is not clear !!!!!! Lines 105, 107, 18, 110, 113, 121, 148, 157, 117, 178, 202 and …..

9-      The size and quality of the figures should be increased.

10-  The authors only reported the results. However, they should talk in detail

11-  Also, to verify the presented results, it is strongly recommended to discuss the results of previous studies.

12-  The conclusion section should be significantly improved and the main results of their studies are discussed.

So, the paper is recommended for publication after addressing the comments carefully with a major revision.

Author Response

Please see the attachment. Thanks!

Reviewer 3 Report (New Reviewer)

The authors have presented an interesting topic that is worth researching. However, the manuscript has errors regarding references in the text, which makes it very difficult to understand the content. In addition, there are language errors in the text, and I recommend checking it with a native speaker. Other minor errors are marked directly in the manuscript.

Round 2

Reviewer 1 Report (Previous Reviewer 2)

All set, The paper is in a good form to be published now

Reviewer 2 Report (New Reviewer)

The authors are appreciated to address comments properly. So, the paper is recommended to publish in the current format

This manuscript is a resubmission of an earlier submission. The following is a list of the peer review reports and author responses from that submission.

Round 1

Reviewer 1 Report

The article deals with the study of the variation of blast furnace slag and silica fume in the preparation of concrete. The rheology is studied by analyzing the variation of these compounds and the ratio of the water content and cementitious material in the samples.

The article is presented in a coherent way between the background, the methodology, the results and the conclusions.

References are adequate and current.

It is necessary to check the relationship of the figures in the text with the number of the title of the figures for their correct description.

The introduction provide sufficient background and include all relevant references. All the cited references are relevant to the research. The research is design appropriate. The methods are adequately described. The results are clearly presented. The conclusions are supported by the results.

Reviewer 2 Report

Well written well presented paper, minor typo errors are there

please have a look on the attached file, the errors are not limited to the marked points so go through all manuscript prior to the final submission
